# Application of Artificial Intelligence in Assessing the Self-Management Practices of Patients with Type 2 Diabetes

**DOI:** 10.3390/healthcare11060903

**Published:** 2023-03-21

**Authors:** Rashid M. Ansari, Mark F. Harris, Hassan Hosseinzadeh, Nicholas Zwar

**Affiliations:** 1School of Public Health and Community Medicine, Faculty of Medicine, University of New South Wales, Sydney, NSW 2052, Australia; 2Centre for Primary Health Care and Equity, University of New South Wales, Sydney, NSW 2052, Australia; 3School of Health and Society, Faculty of Science, Medicine and Health, University of Wollongong, Sydney, NSW 2522, Australia; 4Faculty of Health Sciences and Medicine, Queensland University, Brisbane, QLD 4072, Australia

**Keywords:** Type 2 diabetes, artificial intelligence, self-management, glycemic control HbA1c

## Abstract

The use of Artificial intelligence in healthcare has evolved substantially in recent years. In medical diagnosis, Artificial intelligence algorithms are used to forecast or diagnose a variety of life-threatening illnesses, including breast cancer, diabetes, heart disease, etc. The main objective of this study is to assess self-management practices among patients with type 2 diabetes in rural areas of Pakistan using Artificial intelligence and machine learning algorithms. Of particular note is the assessment of the factors associated with poor self-management activities, such as non-adhering to medications, poor eating habits, lack of physical activities, and poor glycemic control (HbA1c %). The sample of 200 participants was purposefully recruited from the medical clinics in rural areas of Pakistan. The artificial neural network algorithm and logistic regression classification algorithms were used to assess diabetes self-management activities. The diabetes dataset was split 80:20 between training and testing; 80% (160) instances were used for training purposes and 20% (40) instances were used for testing purposes, while the algorithms’ overall performance was measured using a confusion matrix. The current study found that self-management efforts and glycemic control were poor among diabetes patients in rural areas of Pakistan. The logistic regression model performance was evaluated based on the confusion matrix. The accuracy of the training set was 98%, while the test set’s accuracy was 97.5%; each set had a recall rate of 79% and 75%, respectively. The output of the confusion matrix showed that only 11 out of 200 patients were correctly assessed/classified as meeting diabetes self-management targets based on the values of HbA1c < 7%. We added a wide range of neurons (32 to 128) in the hidden layers to train the artificial neural network models. The results showed that the model with three hidden layers and Adam’s optimisation function achieved 98% accuracy on the validation set. This study has assessed the factors associated with poor self-management activities among patients with type 2 diabetes in rural areas of Pakistan. The use of a wide range of neurons in the hidden layers to train the artificial neural network models improved outcomes, confirming the model’s effectiveness and efficiency in assessing diabetes self-management activities from the required data attributes.

## 1. Introduction

Diabetes mellitus is one of the leading causes of chronic health problems globally [1]. The International Diabetes Federation (IDF) estimated a worldwide population of 463 million diabetics in 2019 [2], with type 2 diabetes being more prevalent in adults aged 40–59 [2,3]. With a diabetes incidence rate of 19.9% among people aged 20–79, Pakistan is among the world’s top-10 countries for diabetes cases [2].

Type 2 diabetes is a significant public health issue in Pakistan, particularly among the population aged 40–60. This demographic group is at high risk for type 2 diabetes because its members are more likely to be overweight or obese, physically inactive, and have unhealthy eating habits [1,2]. In addition, the social and health disparities of the region’s population [3,4,5,6] contribute to the high prevalence of diabetes and obesity.

The literature considers self-management to be the cornerstone of diabetes care [7]. Several studies [8,9,10] have highlighted the importance of diabetes self-management and its association with improved diabetes knowledge, responsible behaviour of patients towards their disease, and improved clinical outcomes.

Diabetes self-management (DSM) plays an essential role in controlling and preventing the disease’s complications. Nonetheless, patients often do not adhere to self-management recommendations [1,2,3], which is extremely concerning. In the middle-aged population of Pakistan, adherence to recommendations and barriers poses significant challenges [4,5,6] due to unhealthy eating patterns and lack of physical activity [6].

In diabetes self-management, AI-based techniques are incorporated into patient self-management tools, clinician tools, and healthcare systems. It has been demonstrated that AI-powered solutions have a significant impact on patient comorbidities, lifestyle choices, and healthcare center visits, both in terms of frequency and duration [11].

In the case of diabetes self-management, m-health devices already enabled patients to track and collect information regarding their blood glucose, diet, and exercise. Machine learning can now be applied to patient data to generate data-driven, patient-specific interventions [12].

As a result, machine learning has the potential to empower patients by providing them with otherwise-unavailable information, assisting them in making data-driven decisions about their health, and nudging them towards adopting healthier lifestyle habits [13].

Data-driven AI applications, especially in healthcare, have revolutionised medical research in several areas, such as disease diagnosis, image processing, and disease prediction. Deep learning models, recurrent neural networks, and genetic algorithms play an important role in Artificial Intelligence applications. AI is ideal for detecting, analyzing, and predicting heart disease [14], diabetes complications [15], breast cancer [16], hepatitis B [17], and COVID-19 severity [18].

In this article, the prediction paradigm for successfully assessing diabetes self-management is considered using a dataset of 200 middle-aged type 2 diabetes patients in rural Pakistan. Typical regression models may provide a solution but assume statistical independence and interdependencies of input and output variables, homogeneity of continuation, and external factors. However, intricate physiological traits usually offend or discredit these assumptions.

Models are being designed to improve diabetes self-management. Various models and programmes have been established to improve medical research, but their testing is incomplete and inadequate, making them challenging to perform and manage. Since AI has been used in medical research and analytical investigations, this research presents a well-organised and developed artificial neural network model that assesses type 2 diabetes self-management activities and behaviours.

This work applies AI models in the diagnostic strategy to measure diabetes self-management behaviours by selecting criteria related to self-management. AI can create and apply such models, which are more useful, efficient, and effective in numerous medical disciplines, such as analysis, diagnosis, and prediction; this development could help professionals and patients alike [19]. ANNs mathematically represent the human brain system, showing the power of training and generalisation. Most ANN approaches use nonlinear functions with complicated or unknown input feature links [19].

A neural network is composed of neuron layers. Each neuron in the ANN model is directly linked to neurons in the other layers via weighted values [19,20]. The weighted permutation of multiple input signals that may contain different computations influences the input and output of each neuron. These neurons determine the threshold value by applying the transfer function to inputs with weights. If the threshold is exceeded, the activation function transmits the signal to the following neuron. Important ANN functions include prediction, perception classification, pattern recognition, and training [20]. 

However, ANN development for medical applications, such as classifications, clustering, data optimisation, and input-based prediction, is ongoing; therefore, it is important to understand that, when they present predictions, perception classifications, and pattern recognitions along with training [20,21], ANN models include input layers, hidden layers, output layers, neurons, and their interactions. Few features hinder training, but several reduce network processing power [21]. This article used ANN and machine learning algorithms such as logistic regression to choose attributes, evaluate data, and assess diabetes self-management activities.

## 2. Material and Method

### 2.1. Dataset Description

Al-Rehman Hospital’s primary healthcare and diabetes management clinic in Abbottabad, Pakistan, recruited 200 patients with diabetes (250 patients were approached). This group comprised poorly managed type 2 diabetes patients aged 40–60 years old (Table 1). This study included patients with Hemoglobin (HbA1c) > 7% (Hemoglobin is a simple blood test that measures average blood sugar levels over the past three months) and excluded those with liver, renal, or thyroid issues. All 200 patients gave informed consent and completed a questionnaire thereafter.

### 2.2. Data Visualization

Figure 1 shows distribution of the variable “Age_year”. The minimum value is 40 years and maximum is 65 years. The verticle line divides the surface area between Q2 quartile (52 years) and Q3 quartile (56 years). Figure 2 provides dustribution of the variable “Body mass index” with minimum value of 16.9 kg/m^2^ and maximum value of 56.5 kg/m^2^. Figure 3 shows dustribution of the variable “DiabetesTime” with minimum and maximum values bewteen 2 years to 13 years. Figure 4 shows the distribution of the variable HbA1c (%) with minimum and maximum values between 6.8% to 13.2%. Figure 5 displays the distribution of the variable “Execise” with minimumand maximum values between 3 to 5 dyas/week. Figure 6 shows the distribution of variable “income”.

### 2.3. Aims and Objective

The main objective of this work was to prepare and carry out diabetes self-management (DSM) assessments for patients with type 2 diabetes in rural areas of Pakistan. The simplified proposed modelling approach is shown in Figure 7. 

The raw dataset includes the input variables used in the main algorithm, such as age (years), BMI (kg/m^2^), exercise, diet, blood glucose testing, medication, formal education, diabetes duration (time), and HbA1c levels (%). The outcome variable is diabetes self-management (DSM). DSM is a function of HbA1c % in the analysis. The lower the levels of HbA1c%, the better the DSM activities. We excluded HbA1c levels (%) as an input variable in our analysis to avoid collinearity since DSM is a function of HbA1c. 

### 2.4. Data Pre-Processing

Data pre-processing is an important part of the data analysis; before model evaluation, many strategies could pre-process the dataset [23,24]. In this study, the preprocessed output matched expectations, and pre-processing responses measured learning rate, momentum, and time, fulfilling the acceptance requirement of the transformed data.

The dataset was imbalanced but had no missing values; therefore, we used SMOTE (Synthetic Minority Oversampling Technique) to fix it [25]. In the proposed methodology or modelling approach shown in Figure 7, we normalised the diabetes dataset, and used 80% for validation and training and 20% for testing. Python programming was used to develop the model. We have applied the selected ANN algorithms and optimisation techniques to obtain the best prediction model to assess diabetes self-management activities. 

## 3. Results 

### 3.1. Algorithms Used for Classification

We used AI algorithms to classify the dataset. The logistic regression algorithm was used as a baseline classification algorithm. Logistic regression is the most suitable method for the analysis of binary classification tasks with the high diagnostic ability [14]; ANN was used as the main algorithm and accommodated several features, such as age, exercise, diet, blood glucose testing, formal education, diabetes duration, and HbA1c levels. Logistic regression models have been employed to solve this type of problem and enhance patients’ diabetes self-management assessment [26]. These models’ basis usually includes inference to statistical independence, the interdependencies of their input and output variables, uniformity of continuity, and presence of external variables.

#### Logistic Regression Analysis 

The logistic regression model used is represented as follows:(1)$$log\left(\pi /\left(1-\pi \right)\right)=\alpha +\beta_{1}x_{1}+\beta_{2}x_{2}\ldots \ldots \ldots .+\beta_{k}x_{k}$$
where 

*π* = Probability of response variable or dependent variable*β*_1_ = Log of without exposure variable*X* = Independent variables (*x*_1_, …….., *x_k_*_,_ are different predictors)*β*_2_ = Change in each unit as an increase in the independent variable or exposure var*α* = the intercept alphaLog is a link function

Logistic regression analysis was carried out using Python Programming, splitting the data into the training set (80%) and test set (20%). The logistic regression model’s performance was evaluated based on the confusion matrix. The accuracy of the training set was 98%, while the test set’s accuracy was 97.5%; each set had a recall rate of 79% and 75%, respectively. The confusion matrices for the training and test sets are displayed in Figure 8 and Figure 9, respectively. Figure 10 shows the results of the Receiver Operating Characteristics (ROC) curves for training and test data. The area under the curve (AUC) is 0.96 on training and test sets. 

For the training set, we correctly assessed/classified that the patients do not follow DSM.

TP = True positive = 146—we correctly predicted/assessed/classified that these patients do not meet Diabetes Self-management (DSM) targets.

TN = True negative = 11—we correctly predicted/assessed/classified that these patients meet DSM targets based on the values of HbA1c < 7%.

FP = False positive = 03—we incorrectly predicted/assessed that these patients do not meet DSM targets.

FN = False negative = 0—we incorrectly predicted/assessed that these patients meet DSM targets.

The confusion matrix for the test set output can also be elaborated. 

Logistic Regression model gave a generalised performance on training and test set.ROC-AUC score of 0.96 on training and test set was promising.

### 3.2. Artificial Neural Network

Establishing a neural network model that accurately assesses patient self-management activities was the main objective of this study. In this study, the network was trained using ANN algorithms. We categorised the 200-patient dataset according to the requirements. Training, validation, and test data that were needed for 200 diabetes patients were divided into a ratio of 80:20 between training and testing; 80% (160) of cases were utilised for training, while 40 instances were selected for testing purposes. The performance of algorithms was evaluated using the confusion matrix.

The other AI-based algorithms, such as support vector machine (SVM) and Naïve Bayes, are the algorithms used most frequently in previous studies to predict and evaluate diabetes management practices [25,26,27]. These algorithms find hidden data by balancing processing time and accuracy [26,27].

In this study, the process of using ANN algorithms to forecast, validate, and test the network to improve diabetes patients’ self-management is displayed in Figure 11. The framework requires the network to collect 200 diabetes patients’ self-management data. Due to noise or null data, various features (diet, exercise, glucose testing, age, formal education, diabetes duration, HbA1c levels, etc.) in the dataset may confuse the results. To minimise errors, we carefully selected these features using the data-pre-processing. 

ANN architecture varies between classifiers, exhibiting underlying algorithm parameters that are dependent on the classifier that is required to train the network. The ANN structure contained an input layer and three hidden layers. Each hidden layer was equipped with an activation function and neurons. Similarly, the second and third hidden layers were applied with different neurons. Finally, we have the output layer, which had only one neuron. Specific applications were also applied to incorporate the optimisation techniques when developing the model within the framework of ANN. 

### 3.3. Artificial Neural Network Models 

#### 3.3.1. ANN Model_1 with SGD Optimiser 

ANN Model_1 was developed using the SGD (Stochastic Gradient Descent) optimiser technique [29,30].

Gradient descent is a well-known optimisation strategy in the field of machine learning and deep learning; the ANN model_1 optimiser follows this data optimisation approach [29]. Instead of consuming the entire dataset in each iteration, the SGD optimiser randomly selects a small subset of samples [30].

We employed the ReLU activation function and 128 neurons in the first hidden layer. With the addition of non-linearity provided by a rectified linear unit (ReLU), we used a deep learning model to avoid the problem of vanishing gradients, ensuring that the positive half of the argument was properly interpreted.

In the second layer, we added 64 neurons with the ReLU activation function. The output layer had one neuron as well as sigmoid as an activation function. Bu having the activation function of a neuron as a sigmoid function, we ensured that the output of this unit consistently falls within the range of 0 and 1, regardless of the state of the neuron. In addition, because the sigmoid is a non-linear function, the output of this unit was be a non-linear function of the weighted sum of the inputs. The accuracy was 90% on the model evaluation of the test data. 

Though the model was overfitting, the training loss was smooth; overall, it decreased in correlation with an increase in the epochs (Figure 12). The confusion matrix (Figure 13) of the model shows that only 36 patients met the DSM targets.

The confusion matrix in Figure 10 shows that 36 patients with type 2 diabetes were correctly classified by the neural network classifier as following DSM targets. None of the type 2 diabetes patients was misclassified by the neural network classifier. 

#### 3.3.2. ANN Model_2 with Adam Optimiser 

The Adam optimiser technique was used to create and use ANN model_2 (Adaptive Moment Estimation). This is an efficient method for stochastic optimisation that only needs first-order gradients and does not need much memory. Estimates of the first and second moments of the gradients are used to calculate the individual adaptive learning rates for each parameter [31].

We added 128 neurons in the first hidden layer and used the ReLU activation function. In the second layer, we added 64 neurons with the ReLU activation function; in the third hidden layer, we added 32 neurons. The output layer contained only one neuron and used sigmoid as an activation function. The accuracy was 100% on the model evaluation of the training and test data. The model was overfitting (as may be observed from Figure 14), but the model accuracy was promising (as shown in Figure 15). 

Table 2 provides a performance comparison for different classification techniques. 

The ANN model using Adams’ optimiser outperformed all other classification and optimisation techniques. The main reason for this outcome is that, similar to the RMSpropr optimiser, the Adam optimiser uses squared gradients to scale the learning rate; it also takes advantage of momentum by moving the average of the gradient in the same way as the SGD optimiser [32]. The criteria of comparison were set to obtain the high score of Recall of the model. The higher the score of Recall, the lower the probability of false negatives. Other scores, such as accuracy and F1, were also considered for comparison purposes.

## 4. Discussions

We used the SGD optimiser technique in this study, which has proved itself as an efficient and effective optimisation method central to many machine learning solutions, such as recent advances in deep learning [33,34]. The accuracy of the ANN model using SGD optimiser was 90% on the model evaluation of the test data. The model predicted/assessed by the use of a confusion matrix that only 36 patients met the diabetes self-management targets. 

In this study, we also employed the Adam optimiser, a method for efficient stochastic optimisation that requires first-order gradients and minimal memory. Using estimations for the first and second moments of the gradients, the approach computes individual adaptive learning rates for various parameters. Our method combined the benefits of two prominent methods within Adam optimisation in online and non-stationary environments [35,36]., The ANN model utilizing the Adam optimiser to examine training and test data with 100% accuracy. 

In this study, AI evaluated type 2 diabetes patients against a four-part diabetes self-management criteria: the practice of diet control, regular physical activity, medication adherence, and glucose monitoring (keeping HbA1c < 7%) [10]. DSM is key to ensuring effective control of serum glucose, which reduces the development of diabetes-related comorbidities [37].

AI analyses revealed the majority of the study’s participants complied with taking medication prescribed by their physician. This high rate of medication adherence was also observed in other studies carried out in Pakistan by Khattab et al. [37] and Ahmad et al. [38]. The study conducted on the US population revealed that just 64% of patients complied with medication adherence [39]. While medication adherence is associated with effective diabetes self-management [40], this study’s high medication compliance rate relative to other DSM behaviours suggested that the majority of patients with type 2 diabetes chose to take medicines rather than adjust their behaviour. This behaviour is a major obstacle to satisfying the diabetes self-management requirements and maintaining a healthy lifestyle. 

The application of AI revealed that diet control was another important feature. This assessment was in agreement with a qualitative study carried out by Ansari et al. [41], which showed that a very low percentage of participants practiced diet control [42]. Other previous research [43,44] identified lack of motivation, the frequency of social meetings, and the time and energy required for meal preparation as factors that impeded diet control. A previous study also demonstrated that counselling on diet control would improve respondents’ comprehension of its significance, resulting in a significant decrease in total HbA1c levels and BMI [29]. 

Few participants engaged in 30 min of physical activities for at least five days per week, as indicated by Jafar et al. [5] and Ansari et al. [6]. However, research conducted in the United States recorded a somewhat greater prevalence of physical activity [45]. The obstacles preventing patients with type 2 diabetes from engaging in physical exercise may include inclement weather, such as hot or rainy days, and staying at home due to a lack of available walking space. 

As nearly half of the respondents were over the 60 years old, many type 2 diabetes patients may not be able to conduct the suggested regular exercise due to poor health; age may also contribute to this low performance. 

The ANN model used a confusion matrix to predict/assess that only 36 patients met the DSM targets. This shows that blood glucose monitoring was not practiced regularly by the participants. The patients with uncontrolled glucose levels (HbA1c > 7%) were less likely to undertake appropriate diabetes care activities than those with controlled glucose levels (*p* = 0.050). The main reason may be the high cost of glucose testing strips, which is beyond the reach of participants with low incomes. 

Longer duration of diabetes was associated with poor glycemic control, though this result was not statistically significant (*p* = 0.422). This result is in agreement with other studies that reported a similar association between long-duration diabetes and poor glycemic control [46]. This relationship may stem from the progressive impairment of insulin secretion with time due to β-cell failure, which does not respond to diet or other oral agents [46]. The other independent variables related to patients’ characteristics, such as age, formal education, and body mass index, had little impact on their diabetes self-management activities (*p* > 0.05). However, the lack of a correlation between glycemic control and age is not consistent with the findings in other studies, which reported that younger age was associated with poor glycemic control [45,46]. 

### 4.1. Clinical Perspectives

Data-driven precision care by AI will usher in a new era in the treatment of diabetes.The study has altered diabetes prognosis and self-management, which may contribute to reducing the worldwide prevalence of type 2 diabetes.Predictive risk stratification of populations, improved decision-making, and self-management are all made possible by artificial neural networks and machine learning.This research study will provide benefits to healthcare professionals in decision-making and remote monitoring of diabetes self-management activities. The precise AI application will contribute to reducing the prevalence of complications from type 2 diabetes. The AI applications will be deployed in other rural areas of Pakistan and may be extended to other countries in the sub-continent.

### 4.2. Strengths and Limitations

The strength of this study is that by assessing diabetes self-management activities, AI will guide patients to improve their diabetes self-care and help health professionals in decision making and remotely monitoring patients’ activities.The limitation of this study is that AI poses a risk of de-skilling general practitioners due to their dependency on it. This may result in a vicious circle of inaccuracy because AI requires periodic refining by specialists [47].The ongoing cost, availability, and implementation of AI are obstacles to the use of AI in rural areas of Pakistan. Interoperability has been cited as a typical barrier to adopting devices and apps in diabetes self-management [47].The other limitation is the relatively small sample size of just 200 patients with type 2 diabetes. In diabetes self-management, a recurring difficulty is the dearth of data needed to develop rational and precise algorithms. The results of this study may not be generalised to other regions. The longer the length of future studies’ training sets, the more accurate and applicable to other regions their results will be. For digital applications to design meaningful solutions, datasets will need to be increasingly developed and organised. Concerns over security, data privacy, and regulatory issues hinder the implementation of technology in diabetes self-management.Using retrospective data, current AI models and applications in diabetes treatment have been validated. The prospective confirmation of these technological breakthroughs has the potential to automate diabetes care [48]. To include digital biomarkers and data from applications, monitors, and activity trackers, endpoints in clinical research will need to be reformulated.

## 5. Conclusions

This study assessed the factors associated with poor diabetes self-management activities among type 2 diabetes patients in rural areas of Pakistan. The use of a wide range of neurons in the hidden layers to train the artificial neural network models improved outcomes, confirming the model’s effectiveness and efficiency in assessing diabetes self-management activities. The optimization techniques used in ANN models identified four important features related to patients’ self-management activities. AI application revealed that the majority of diabetes patients heavily relied on medication adherence to manage their disease, rather than adjusting their self-management behaviour. Resultantly, it will remain challenging for healthcare professionals to encourage rural patients to adopt healthy lifestyles. 

In future studies, AI may be extended to develop specific web-based applications to facilitate patients’ self-management activities; these applications’ features map include advice on diet control, planning physical activity routines, and glucose monitoring and control.

## Figures and Tables

**Figure 1 healthcare-11-00903-f001:**
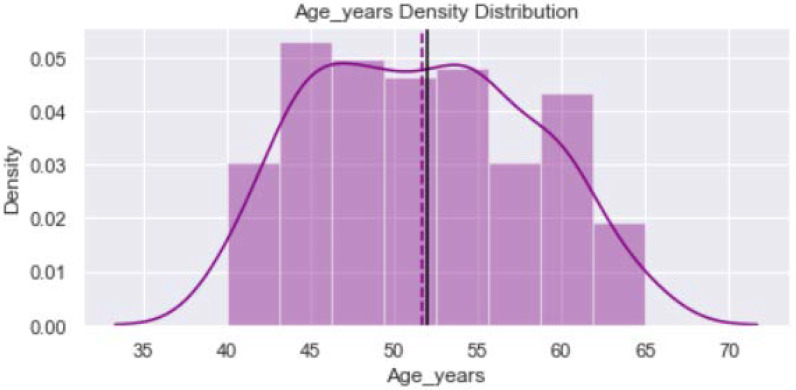
The distribution of variable age.

**Figure 2 healthcare-11-00903-f002:**
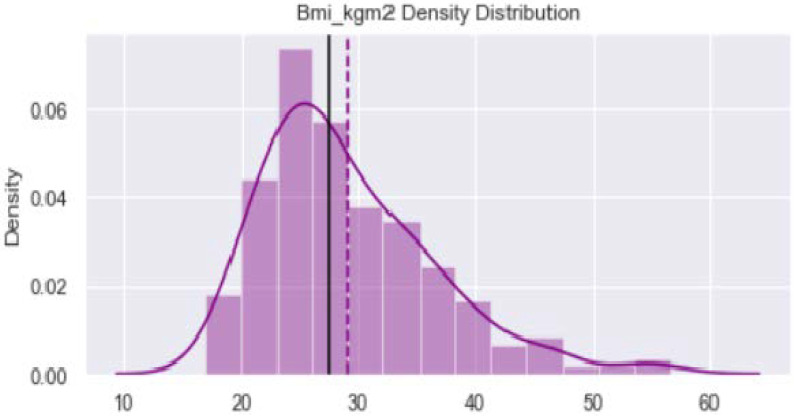
The distribution of variable body mass index.

**Figure 3 healthcare-11-00903-f003:**
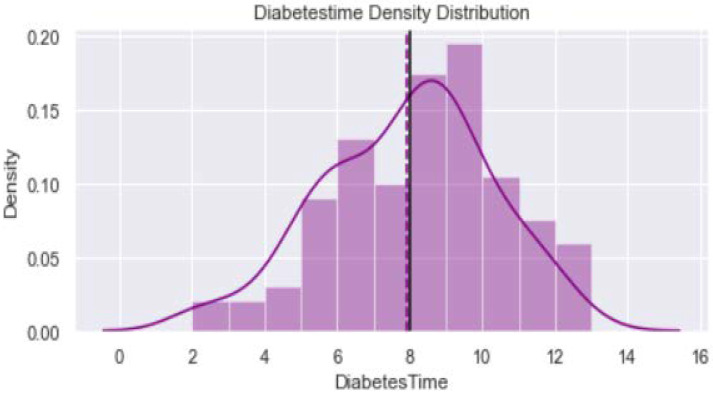
The distribution of variable DiabetesTime.

**Figure 4 healthcare-11-00903-f004:**
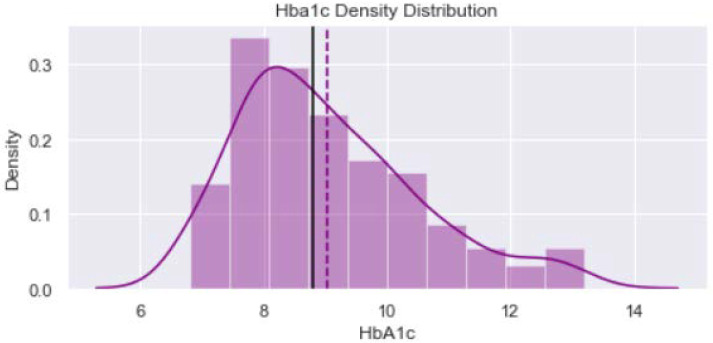
The distribution of variable HbA1c.

**Figure 5 healthcare-11-00903-f005:**
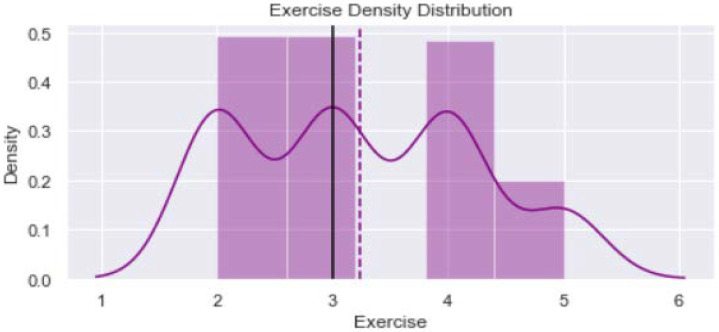
The distribution of variable exercise.

**Figure 6 healthcare-11-00903-f006:**
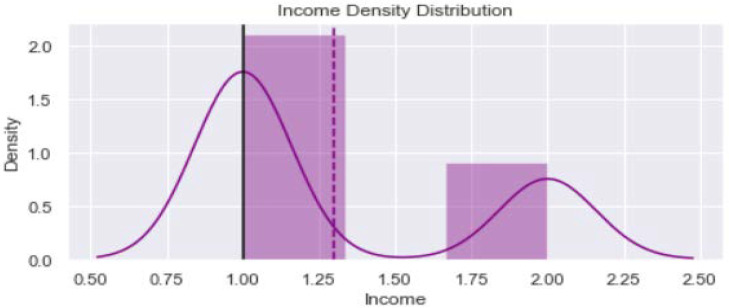
The distribution of variable income.

**Figure 7 healthcare-11-00903-f007:**
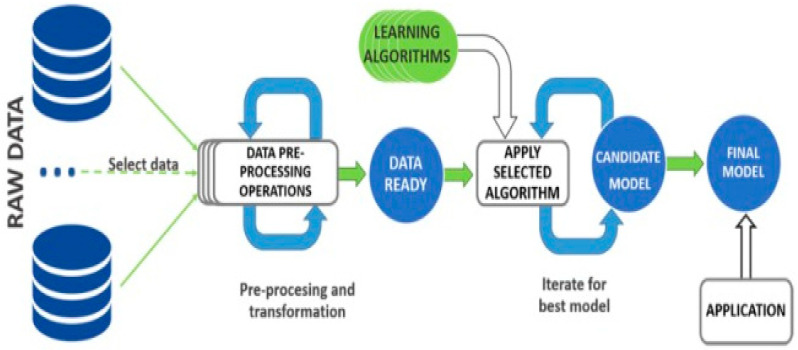
Proposed modelling approach—modified from the source: Contreras & Vehi [22].

**Figure 8 healthcare-11-00903-f008:**
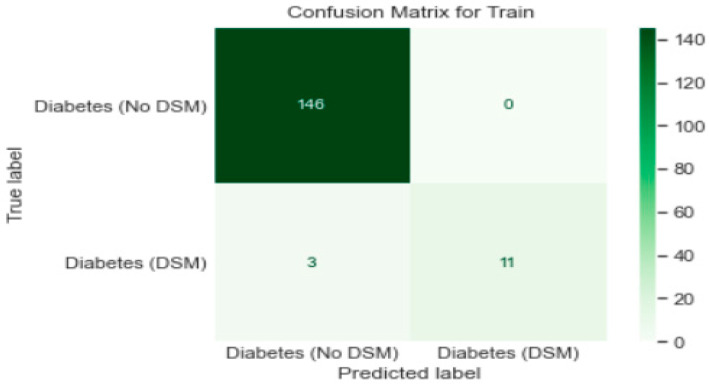
Confusion Matrix for training sets for patients with diabetes (*n* = 200).

**Figure 9 healthcare-11-00903-f009:**
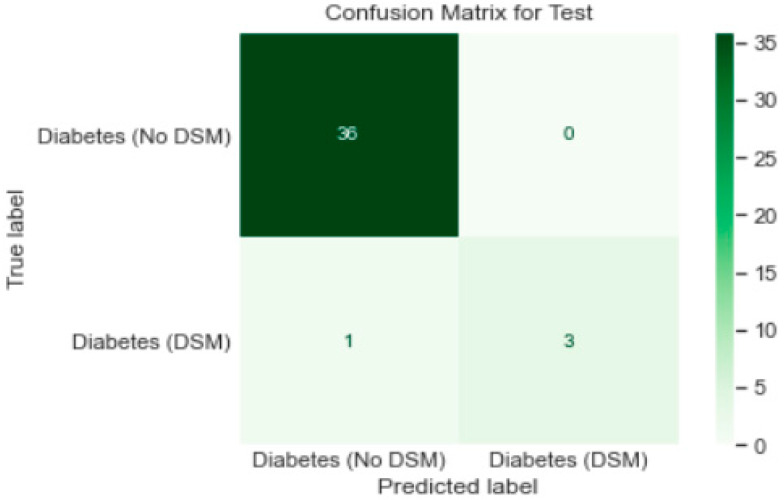
Confusion matrix for test sets for patients with diabetes (*n* = 200).

**Figure 10 healthcare-11-00903-f010:**
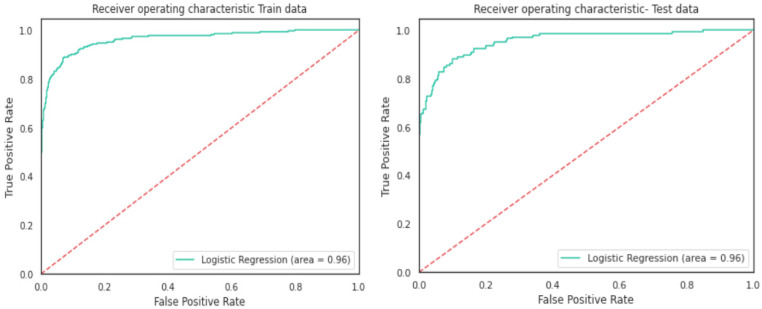
Receiver operating characteristic (ROC) curves for training and test data.

**Figure 11 healthcare-11-00903-f011:**
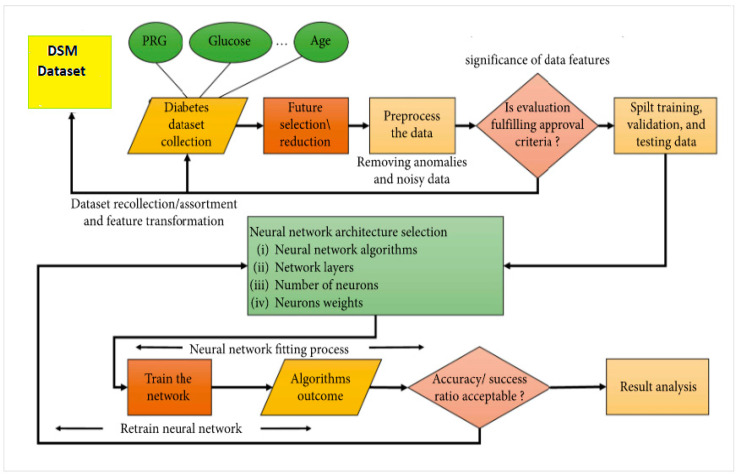
The framework of diabetes self-management (DSM) prediction using the artificial neural network (ANN)—modified from the source: Bukhari et al. [28].

**Figure 12 healthcare-11-00903-f012:**
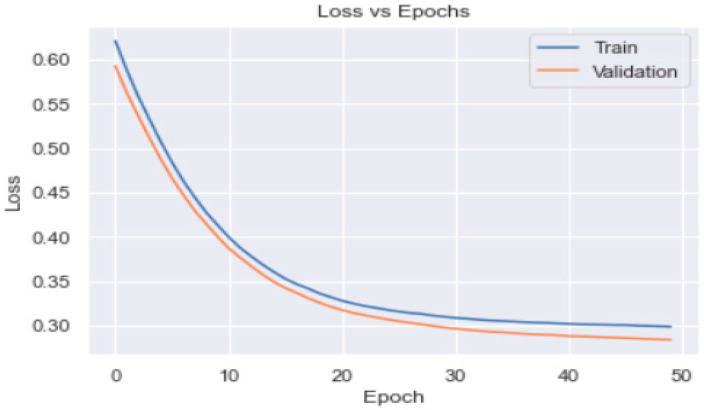
Training and validation loss profile of the ANN model_1.

**Figure 13 healthcare-11-00903-f013:**
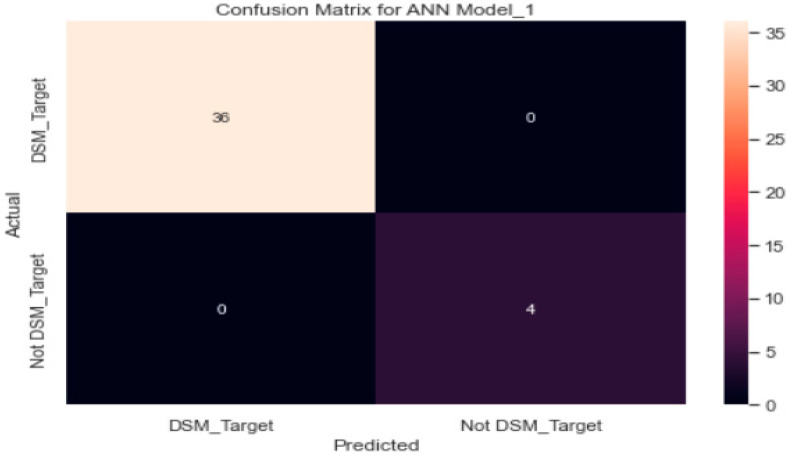
Confusion matrix for the artificial neural network algorithm (model_1).

**Figure 14 healthcare-11-00903-f014:**
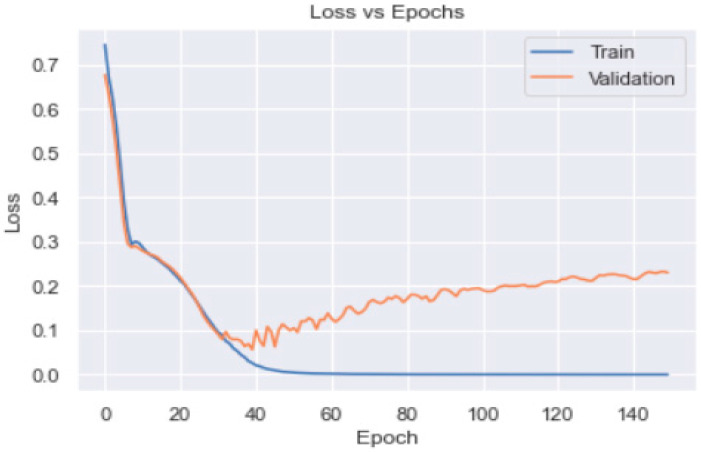
Training and validation loss profile of the ANN model_2.

**Figure 15 healthcare-11-00903-f015:**
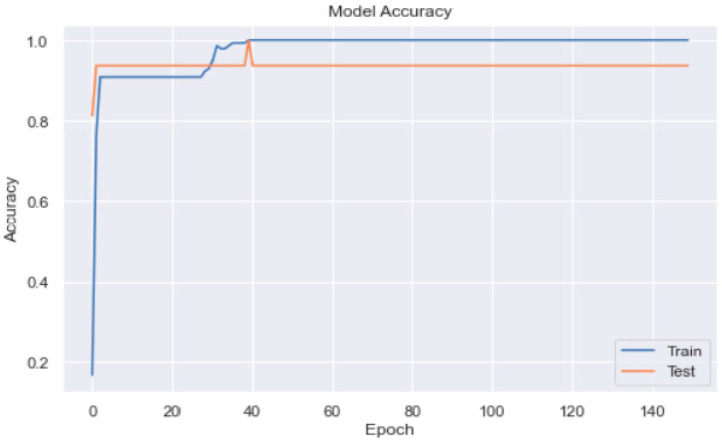
The ANN model_2 accuracy with three hidden layers.

**Table 1 healthcare-11-00903-t001:** Patients’ characteristics and their association with glycemic control (*n* = 200).

Parameters	Male (*n*)	Female (*n*)	Mean ± SD	*p*-Value	Total	
**Age** (in years)	51	53	51.40 ± 6.42	0.25	52	
<60 years	85	87			172	
≥60 years	15	13			28	
**Diabetes Patients**	100	100		0.20	200	
**Marital Status**						
Single	15	5			20	
Married	75	85			160	
Divorced	10	2			12	
Widowed	0	8			8	
**Education**						
<grade 9	16	50			66	
High School	65	40			105	
College degree	10	7			17	
Professional	9	3		0.70	12	
**Employment**						
Full/part time	75	65		0.05	140	
Unemployed	10	35			45	
Retired	15	0			15	
**Diabetes Duration**						
<8 years	36	42	7.72 ± 2.38		78	
≥8 years	64	58	8.1 ± 2.30	0.048	122	
**HbA1c (%)**						
Uncontrolled (>7%)	91	91	9.03 ± 1.52	0.050	182	
Controlled (≤7%)	9	9			18	

Notes: *n* = number of patients; SD = standard deviation; *p*-values are two-tailed *t*-test values.

**Table 2 healthcare-11-00903-t002:** Performance of different classification techniques.

Classification	Accuracy	Precision	Recall	F1 Score
Logistic regression	98%	97.3%	79%	88%
Artificial neural network				
SGD optimizer	90%	74%	74%	74%
Adam optimiser	100%	100%	100%	100%
RMSprop	84%	82%	81%	82%

## Data Availability

The datasets used and or/analysed during the current study are available from the corresponding author upon reasonable request.

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
