# Peer review of "Application of Artificial Intelligence in Assessing the Self-Management Practices of Patients with Type 2 Diabetes"

_healthcare, 2023, doi:10.3390/healthcare11060903_

Round 1

Reviewer 1 Report

A pdf with corrections is added to this review

NOTES

·         English generally sound

·         Nearly 50 revisions needed as in the attached pdf

·         Should headings be numbered?

·         Needs careful proofreading and rewrite in parts – a few typos and grammatical errors

·         Acronyms such as AI, ANN, DSM should be used consistently – just need to define at first use then use the acronym

·         The small datasets are a concern; a larger study could follow on from this. This is noted in the paper.

·         Interesting comparison between Pakistan and US approaches to DSM. why is this?

·         3 papers by authors in the references

DETAILED COMMENTS ON PAPER

Abstract

·         Should provide a summary of paper.

·         Suggest removing bolded headings and making more general – no need to give detailed results such as The accuracy of the training set was 98% and the test set was 97.5% with a recall rate of 79% 30 and 75% respectively for the training and test sets.

1.       Introduction

·         Adequately introduces the topic

2.       Material and Methods

·         Satisfactory level of detail

3.       Results

·         Logistic regression model – what are key parameters used from table 1 – age, sex, marital status, education, employment, diabetes duration? Which are most significant? Is education more significant than age, for example?

·         A confusion matrix is provided and then a set of ROC curves. It is not clear how these are derived. The confusion matrix gives a single point on the ROC curve. What is varied to get other values on the ROC curve?

·         Note that fig 7 is not referenced

4.       Artificial Neural Network Models

5.       Discussions

·         Should these be subheadings under Discussions with italic font?

o   Clinical Perspectives

o   Strength and Limitations

6.       Conclusions

·         Short but OK

7.       Reference

·         Could add more

Author Response

Reviewer 1 Comments

NOTES

  • English generally sound – Thanks
  • Nearly 50 revisions needed as in the attached pdf

I have followed all the suggestions and modified the text accordingly. We really appreciate the suggestions/corrections of reviewer 1. (Thanks for your great observation).

  • Should headings be numbered?

We have followed the guidelines of the healthcare MDPI journal and modified the text accordingly.

  • Needs careful proofreading and rewriting in parts – a few typos and grammatical errors

We have checked for any typos and grammatical errors.

  • Acronyms such as AI, ANN, and DSM should be used consistently – just need to define at first use then use the acronym

Followed as suggested.

  • The small datasets are a concern; a larger study could follow on from this. This is noted in the paper.

Thanks for the observation. Yes, this is the limitation of our study and a large database will be a better representation of that population.

  • Interesting comparison between Pakistan and US approaches to DSM. why is this?

The main reason for that in Pakistani culture diet control and exercise is considered the most difficult activity to follow (family tradition, places to exercise particularly for women).  AI models also identified that. However, in some studies in the US, participants originally belonged to Pakistan and performed these activities adequately. This makes it very interesting how the environment influences the perception of diabetes self-management.  (This particular aspect was not explored in this article).

        3 papers by authors in the references

The references are important as these references provide more details. For example, the same database was used for Structural Equation Modelling (Reference 17) and the results obtained are in agreement with the current article on AI. In fact, AI provided better insight into Diabetes Self-management assessment.

DETAILED COMMENTS ON PAPER

Abstract

  • Should provide a summary of the paper.
  • Suggest removing bolded headings and making more general – no need to give detailed results such as The accuracy of the training set was 98% and the test set was 97.5% with a recall rate of 79% 30 and 75% respectively for the training and test sets.

We have used the structured abstract format as some of the articles in this journal used it. As suggested, we have removed the bolded heading in the text of the abstract. As recommended, we have removed the detailed results. We thought, maybe these are significant given the analysis of a small sample size.

  1. Introduction
  • Adequately introduces the topic. Thanks
  1. Material and Methods
  • Satisfactory level of detail. Thanks
  1. Results
  • Logistic regression model – what are key parameters used from table 1 – age, sex, marital status, education, employment, diabetes duration? Which are most significant? Is education more significant than age, for example?

We have used age (years), BMI (kg/m2), exercise, diet, blood glucose testing, medication, formal education, diabetes duration (time), and HbA1c levels (%). The outcome variable is Diabetes Self-Management (DSM). DSM is a function of HbA1c % in the analysis. The lower the levels of HbA1c%, the better the DSM activities.

In logistic regression analysis, initially, all these independent variables were used and observed their contribution to the outcome of the model. The variables with p<0.05 (statistically significant) were retained for further analysis. 

  • A confusion matrix is provided and then a set of ROC curves. It is not clear how these are derived. The confusion matrix gives a single point on the ROC curve. What is varied to get other values on the ROC curve?

We have used the same output for ROC Curve from the model which produced a confusing matrix. However, we have drawn these curves with various independent variables using STATA software but did not include them in this article.

Note that fig 7 is not referenced: Figure 7 is now referenced in the text. Thanks for the observation

  1. Artificial Neural Network Models – Thanks but there was no query there.
  2. Discussions
  • Should these be subheadings under Discussions with italic font?

o   Clinical Perspectives

  • Strengths and Limitations

We have provided sub-headings under the main heading “4. Discussions”

4.1 Clinical perspectives

4.2 Strengths and limitations

  1. Conclusions
  • Short but OK. Thanks
  1. Reference

         Could add more.

Yes, we have added more references and extended the text in the Introduction and other sections as well,

Reviewer 2 Report

This research requires many questions to the answered(implemented). In addition, I list issues that must be answered(implemented):

1. You have to specify very carefully the input set of variables upon which predictions are made. (you have mentioned the in-section Results, but they must be also enumerated Objectives section at the beginning).

2. The previous must be also done for the output variable(s).

3. Introduction is quite short, I cannot recommend paper acceptance if less than 10 citations are provided there.

4. Reasonable explanation why logistic regression and ANN were used as classification models must be provided.

5. Links to hospitals wherefrom data was collected must be provided.

6. The dataset upon which computations were done must be provided in form of appendix in order to be confirmed obtained results.

7. Reference regarding SMOTE, when first mentioning must be provided.

8. Unit values for all input variables must be declared.

9. The architecture of ANN or part of it must be depicted in the paper.

10. Reference to Stochastic Gradient Descent method must be provided when first mentioning.

11. Reference to Adam optimizer technique must be provided.

12. Pakistan patients participated in the study, do you expect to get the same results if data was taken from other region and explain how the length of training set

    will impact the accuracy of classification.

13. Emphasize the benefit of your research from medical aspect, stress its application, eventual application deployment in the future.

Author Response

Reviewer 2 Comments

This research requires many questions to the answered(implemented). In addition, I list issues that must be answered(implemented):

  1. You have to specify very carefully the input set of variables upon which predictions are made. (you have mentioned the in-section Results, but they must be also enumerated Objectives section at the beginning).

It is specified in the text related to “Objectives” as suggested. Thanks for this observation

The raw dataset includes the input variables used in the main algorithm with features such as age (years), BMI (kg/m2), exercise, diet, blood glucose testing, medication, formal education, diabetes duration (time), and HbA1c levels (%). The outcome variable is Diabetes Self-Management (DSM). DSM is a function of HbA1c % in the analysis. The lower the levels of HbA1c%, the better the DSM activities.

  1. The previous must also be done for the output variable(s).

The output variable (outcome variable: DSM) has been mentioned in the text as well.

  1. Introduction is quite short; I cannot recommend paper acceptance if less than 10 citations are provided there.

Thanks for the observation. We have extended the introduction section and the number of citations is more than 10 now as recommended. 

  1. Reasonable explanation why logistic regression and ANN were used as classification models must be provided.

The logistic regression algorithm was used as a baseline classification algorithm because it is an adequate approach to be used as a baseline classification model and ANN was used as the main algorithm.

  1. Links to hospitals where from data was collected must be provided.

We could not establish any online links to the hospital database. In rural areas, there was very poor internet access and the server used had some limitations.

  1. The dataset upon which computations were done must be provided in form of an appendix in order to be confirmed obtained results.

The dataset of diabetes patients is available with a reasonable justification upon request.

  1. Reference regarding SMOTE, when first mentioned must be provided.

Reference [25] related to SMOTE has been provided in the text and in the list of references.

[25] Chawla, NV, Bowyer, KW, Hall, LO, et al. SMOTE: Synthetic Minority over-sampling Technique. Journal of Artificial Intelligence Research 16(2002), 321-357.

  1. Unit values for all input variables must be declared.

The unit values of all the input variables have been defined in the text as well as in Table 1.

  1. The architecture of ANN or part of it must be depicted in the paper.

Thanks for this observation. Yes, we have shown the architecture of ANN in Figure 8 and defined its applications in section 3.3 ANN Models. 

  1. Reference to the Stochastic Gradient Descent method must be provided when first mentioned.

 References have been provided. (Ref: 30 and 31)

  1. Reference to the Adam optimizer technique must be provided. Yes. Agree

[32] Diederik, P, & Kingma and Jimy bei Ba. Adam: A method for Stochastic optimization. 2014, arXiv: 1412-6980.

  1. Pakistan patients participated in the study, do you expect to get the same results if data was taken from other region and explain how the length of training set will impact the accuracy of classification.

As mentioned in the limitation section, we had a relatively small sample size of just 200 patients with type 2 diabetes. In diabetes self-management, a recurring difficulty is the dearth of data needed to develop rational and precise algorithms. The results of this study may not be generalized to other regions but the larger the length of the training set, the better will be the accuracy and applicability to other regions.

  1. Emphasize the benefit of your research from medical aspect, stress its application, eventual application deployment in the future.

 This research study will provide benefits to healthcare professionals in decision-making and remote monitoring of diabetes self-management activities. The precise AI application will contribute to reducing the prevalence of type 2 diabetes. The AI applications will be deployed in rural areas of Pakistan and may be extended to other countries in the sub-continent.

  • We have included that in the “Clinical Perspectives” section with some modifications.
  • Thanks for your valuable comments on items 12 and 13.

Reviewer 3 Report

The topic of the article is interesting, and the results of such an approach potentially promise benefits for practice.

From my point of view, articles intended for health practice must have a structure that is clearly recognizable: objective, methods, data, results, discussion and conclusion.

References are cited primarily in the Introduction, description of methods and data, and Discussion (not in Results).  References are expected to be recent. In this article, there are less than a quarter of references from the last 5 years (and there are certainly more recent references).

About the data: The data sample consists of 200 patients who were divided into learning and testing subsamples in the ratio of 80:20. How? It should be described according to which criteria the patient was classified into a particular subsample.

Furthermore, a series of patient data is listed, but it is not clear which of these data are processed, for example, what are predictors and what are criteria in logistic regression. BMI is also mentioned - but what about it? There are also mentions of adherence, diet, physical activity, etc. These data from the sample should also be described (in case they have been processed). In the data visualization, it is not clear what the vertical lines mean. Does it have anything to do with data preprocessing?

"Playing" with neurons in ANN is interesting. But how can it be concluded from the various ANN models which patient behaviors affect the result of regulating diabetes? It is expected that the article will answer these questions.

Conclusion should be based on results described in Results paragraph.

Author Response

Reviewer 3 Comments

The topic of the article is interesting, and the results of such an approach potentially promise benefits for the practice.

Thanks for your comments

From my point of view, articles intended for health practice must have a structure that is clearly recognizable: objective, methods, data, results, discussion, and conclusion.

We have followed the MDPI healthcare journal guidelines and your suggestions. Thanks

References are cited primarily in the Introduction, description of methods and data, and Discussion (not in Results).  References are expected to be recent. In this article, there are less than a quarter of references from the last 5 years (and there are certainly more recent references).

We have provided an extension to the Introduction section with new references and modified the results section to compare the results with other publications in this area.

About the data: The data sample consists of 200 patients who were divided into learning and testing subsamples in the ratio of 80:20. How? It should be described according to which criteria the patient was classified into a particular subsample.

Python program provides options to split the data into various sub-samples. Since the sample size was not large, we have divided that into 80:20. The split retains the same composition of patients as before. In cases when the sample size is large, the division is 70:30.

Furthermore, a series of patient data is listed, but it is not clear which of these data are processed, for example, what are predictors and what are criteria in logistic regression. BMI is also mentioned - but what about it? There are also mentions of adherence, diet, physical activity, etc. These data from the sample should also be described (in case they have been processed). In the data visualization, it is not clear what vertical lines mean. Does it have anything to do with data preprocessing?

The raw dataset includes the input variables used in the main algorithm with features such as age (years), BMI (kg/m2), exercise, diet, blood glucose testing, medication, formal education, diabetes duration (time), and HbA1c levels (%). The outcome variable is Diabetes Self-Management (DSM). DSM is a function of HbA1c % in the analysis. The lower the levels of HbA1c%, the better the DSM activities.

In logistic regression analysis, initially, all these independent variables were used and observed their contribution to the outcome of the model. The variables with p<0.05 (statistically significant) were retained for further analysis.  This vertical line on the graph is part of the data visualization and does not contribute to our analysis. It appears to divide the graph into Q2 and Q3 quartiles (Ref: variable Age)

"Playing" with neurons in ANN is interesting. But how can it be concluded from the various ANN models which patient behaviors affect the result of regulating diabetes? It is expected that the article will answer these questions.

Thanks for this observation. Yes, AI models provided insight into the four important DSM activities such as diet control, physical activity (exercise), medication adherence, and glucose monitoring (keeping HbA1c <7%). The two ANN models provided information on which activity was more or less used by patients and its overall impact on diabetes self-management. This has been explained in the text under the ANN models section.

The conclusion should be based on the results described in the Results paragraph. Well noted

Round 2

Reviewer 2 Report

The quality of the paper was partially improved.

However, there is additional work to be done.

All questions bellow must be mandatory answered and strictly implemented as requested, before recommending paper acceptance.

Answer and implement strictly as requested:

Question 1:

_________________________________________________________________________________

Please find the following sentence in your section Introduction:

AI is ideal for detecting, analyzing, and predicting heart disease, diabetes complications, breast cancer, hepatitis B, and Covid-19 severity [14].

When mentioned that AI can be used for medical diagnosis purpose, you must also cite relevant and current research in the specific topic (published 2023 or 2022). AI models which are based on logistic regression that is the core model in your research can only be accepted.

I provide you two references that make use of logistic regression for heart disease (Stojanov et al., 2023), diabetes (Edlitz & Segal, 2022) and in the same fashion you will have to find 2022/2023 research papers that make use of logistic regression for diagnosis of breast cancer (Ref.) and hepatitis B (Ref.).

The final sentence should look like (formatted according to the Healthcare citation style):

AI is ideal for detecting, analyzing, and predicting heart disease (Stojanov et al., 2023), diabetes complications (Edlitz & Segal, 2022), breast cancer (Ref.), hepatitis B (Ref.), and Covid-19 severity [14].

Incorporate references (Stojanov et al., 2023), (Edlitz & Segal, 2022) in your research and add two more, one that cites AI breast cancer diagnosis using logistic regression and other hepatitis B.

Stojanov, D., Lazarova, E., Veljkova, E., Rubartelli, P. and Giacomini, M., 2023. Predicting the outcome of heart failure against chronic-ischemic heart disease in elderly population–Machine learning approach based on logistic regression, case to Villa Scassi hospital Genoa, Italy. Journal of King Saud University-Science35(3), p.102573.

https://doi.org/10.1016/j.jksus.2023.102573

Edlitz, Y. and Segal, E., 2022. Prediction of type 2 diabetes mellitus onset using logistic regression-based scorecards. Elife11, p.e71862.

https://doi.org/10.7554/eLife.71862

__________________________________________________________________________________

Question 2:

The answer of question 4. is not quite correct and more specific answer must be provided.

Question 4. Reasonable explanation why logistic regression and ANN were used as classification models must be provided.

Your answer:

The logistic regression algorithm was used as a baseline classification algorithm because it is an adequate approach to be used as a baseline classification model and ANN was used as the main algorithm.

The answer of this question is:

“Logistic regression is the most suitable method for analysis of binary classification tasks with high diagnostic ability.” – paragraph from the research: https://doi.org/10.1016/j.jksus.2023.102573 (Stojanov et al., 2023)

This explanation must be included, by providing the citation also (Stojanov et al., 2023).

_______________________________________________________________

Question 3.

  1. Links to hospitals where from data was collected must be provided.

I can’t accept your answer:

We could not establish any online links to the hospital database. In rural areas, there was very poor internet access and the server used had some limitations.

At least you must provide some kind of link wherefrom data was collected.

Question 4.

  1. The dataset upon which computations were done must be provided in form of an appendix in order to be confirmed obtained results.

Your answer: The dataset of diabetes patients is available with a reasonable justification upon request.

This statement must be included in the paper.

Question 5.

  1. The architecture of ANN or part of it must be depicted in the paper.

Your answer: Thanks for this observation. Yes, we have shown the architecture of ANN in Figure 8 and defined its applications in section 3.3 ANN Models.

Short but detailed description about the ANN architecture must be incorporated in the text.

_________________________________________________________________________

Author Response

Reviewer 2 Comments

Question 1:

Please find the following sentence in your section Introduction:

AI is ideal for detecting, analyzing, and predicting heart disease, diabetes complications, breast cancer, hepatitis B, and Covid-19 severity [14].

When mentioned that AI can be used for medical diagnosis purpose, you must also cite relevant and current research in the specific topic (published 2023 or 2022). AI models which are based on logistic regression that is the core model in your research can only be accepted.

I provide you two references that make use of logistic regression for heart disease (Stojanov et al., 2023), diabetes (Edlitz & Segal, 2022) and in the same fashion you will have to find 2022/2023 research papers that make use of logistic regression for diagnosis of breast cancer (Ref.) and hepatitis B (Ref.).

The final sentence should look like (formatted according to the Healthcare citation style):

AI is ideal for detecting, analyzing, and predicting heart disease (Stojanov et al., 2023), diabetes complications (Edlitz & Segal, 2022), breast cancer (Ref.), hepatitis B (Ref.), and Covid-19 severity [14].

Incorporate references (Stojanov et al., 2023), (Edlitz & Segal, 2022) in your research and add two more, one that cites AI breast cancer diagnosis using logistic regression and other hepatitis B.

Stojanov, D., Lazarova, E., Veljkova, E., Rubartelli, P. and Giacomini, M., 2023. Predicting the outcome of heart failure against chronic-ischemic heart disease in elderly population–Machine learning approach based on logistic regression, case to Villa Scassi hospital Genoa, Italy. Journal of King Saud University-Science35(3), p.102573.

https://doi.org/10.1016/j.jksus.2023.102573

 Edlitz, Y. and Segal, E., 2022. Prediction of type 2 diabetes mellitus onset using logistic regression-based scorecards. Elife11, p.e71862.

https://doi.org/10.7554/eLife.71862

Reply to Q1:

AI is ideal for detecting, analyzing, and predicting heart disease [14], diabetes complications [15], breast cancer [16], hepatitis B [17], and Covid-19 severity [18].

The following references have been added as recommended.

  • Stojanov, D., Lazarova, E., Veljkova, E., et al. Predicting the outcome of heart failure against chronic-ischemic heart disease in elderly population–Machine learning approach based on logistic regression, case to Villa Scassi hospital Genoa, Italy. Journal of King Saud University-Science, 2023, 35(3), p.102573.https://doi.org/10.1016/j.jksus.2023.102573
  • Edlitz, Y. and Segal, E. Prediction of type 2 diabetes mellitus onset using logistic regression-based scorecards. Elife, 2022, 11, p.e71862. https://doi.org/10.7554/eLife.71862
  • Alzubi, A, Najadat, H, Daulat, W et al. Predicting the recurrence of breast cancer using machine learning algorithms. Multimed. Tools Appl. 2021, 80: 13787-13800.doi: 10.1007/s11042-020-10448-w.
  • Elsayed, A, Nassef, A, Dhifullah, M. Diagnosis of Hepatitis Disease with logistic regression and Artificial Neural Networks. Journal of Computer Science. 2020, 16(3), 364-377.doi: 10.3844/jcssp.2020.364.377.
  •  Meraihi, Y, Gabis, AB, Mirjalili, S. et al. Machine Learning-Based Research for COVID-19 Detection, Diagnosis, and Prediction: A Survey. SN COMPUT. SCI. 3, 286 (2022). https://doi.org/10.1007/s42979-022-01184-z

Question 2:

The answer of question 4. is not quite correct and more specific answer must be provided.

Question 4. Reasonable explanation why logistic regression and ANN were used as classification models must be provided.

Your answer:

The logistic regression algorithm was used as a baseline classification algorithm because it is an adequate approach to be used as a baseline classification model and ANN was used as the main algorithm.

The answer of this question is:

“Logistic regression is the most suitable method for analysis of binary classification tasks with high diagnostic ability.” – paragraph from the research: https://doi.org/10.1016/j.jksus.2023.102573 (Stojanov et al., 2023)

This explanation must be included, by providing the citation also (Stojanov et al., 2023).

Reply to Q2.

Thanks for your comments and reply as well. The following statement is included in the text.                     

Logistic regression is the most suitable method for the analysis of binary classification tasks with the high diagnostic ability (Stojanov et al., 2023)

  • Stojanov, D., Lazarova, E., Veljkova, E., et al. Predicting the outcome of heart failure against chronic-ischemic heart disease in elderly population–Machine learning approach based on logistic regression, case to Villa Scassi hospital Genoa, Italy. Journal of King Saud University-Science, 2023, 35(3), p.102573.https://doi.org/10.1016/j.jksus.2023.102573

Question 3.

Links to hospitals where data was collected must be provided.

I can’t accept your answer:

We could not establish any online links to the hospital database. In rural areas, there was very poor internet access and the server used had some limitations.

At least you must provide some kind of link wherefrom data was collected.

Reply to Q3.

There was no link to the data collection from the hospital. The sample of participants was purposively recruited from the medical clinic of Al-Rehman Hospital, Pakistan which provides primary healthcare services, including the management of chronic diseases such as diabetes. Initially, about 250 patients were approached and 200 patients agreed to participate in the study.

The 200 participants recruited were asked to complete the questionnaire following their informed consent. The study was approved by the ethics committee of the University of New South Wales, Australia, and by the Ayub Medical Institution, Pakistan. Most participants completed the questionnaire in the medical clinic waiting room of the facility.

Question 4.

The dataset upon which computations were done must be provided in form of an appendix in order to be confirmed obtained results.

Your answer: The dataset of diabetes patients is available with a reasonable justification upon request.

This statement must be included in the paper.

Reply to Q4.

Please note that we have mentioned that in the text after the conclusion section as follows:

Availability of data and material:

The datasets used and or/analysed during the current study are available from the corresponding author upon reasonable request.

Question 5.

The architecture of ANN or part of it must be depicted in the paper.

Your answer: Thanks for this observation. Yes, we have shown the architecture of ANN in Figure 8 and defined its applications in section 3.3 ANN Models.

Short but detailed description about the ANN architecture must be incorporated in the text.

Reply to Q5.

ANN architecture varies from classifier to classifier, exhibiting the underlying algorithm parameters that are dependent on the classifier that is required to train the network.

ANN structure contained an input layer, three hidden layers, and one output layer. Each hidden layer is equipped with an activation function and neurons. Similarly, the second and third hidden layers were applied with different neurons. Finally, we have the output layer with only one neuron. More details are given with specific applications applied to incorporate the optimization techniques during the model development within the framework of ANN.

Thanks for your excellent comments and suggestions.

Reviewer 3 Report

Different references are cited differently in the article (different referencing styles were used). The method of citation should be standardized and harmonized with the instructions of the journal. 

Author Response

Q: The method of citation should be standardized and harmonized with the instructions of the journal. 

Thanks for the comments. We have followed the instructions and given more attention to consistency in citing the references with the requirement of the MDPI healthcare journal.